# Self-Calibration Method for Circular Encoders Based on Inertia and a Single Read-Head

**DOI:** 10.3390/s24103069

**Published:** 2024-05-11

**Authors:** Xiaoyi Wang, Longyuan Xiao, Kunlei Zheng, Chengxiang Zhao, Mingkang Liu, Tianyang Yao, Dongjie Zhu, Gaojie Liang, Zhaoyao Shi

**Affiliations:** 1School of Mechatronics Engineering, Henan University of Science and Technology, Luoyang 471003, China; xly_yz163@163.com (L.X.); zkl_inbox@163.com (K.Z.); zcx_98@126.com (C.Z.); kklmk163@163.com (M.L.); yty0101@139.com (T.Y.); zdjwsbn@163.com (D.Z.); 2Henan Key Laboratory of Mechanical Design and Transmission System, Henan University of Science and Technology, Luoyang 471003, China; 3Luoyang Transportation Development Center, Luoyang 471000, China; lylgj2012@163.com; 4College of Mechanical & Energy Engineering, Beijing University of Technology, Beijing 100000, China; shizhaoyao@bjut.edu.cn

**Keywords:** circular encoder, single read head, error of angle measurement, self-calibration

## Abstract

This article proposes a new self-calibration method for circular encoders based on inertia and a single read-head. The velocity curves of the circular encoder are fitted with polynomials and, based on the principle of circle closure and the periodicity of the distribution for angle intervals, the proportionality between the theoretical value and the actual value of each angle interval is obtained. In the experimental system constructed, the feasibility of the proposed method was verified through self-calibration experiments, repeatability experiments, and comparative experiments with the time-measurement dynamic reversal (TDR) method. In addition, this article also proposes an iterative method to improve the self-calibration accuracy. Experimental verification was carried out, and the results show that the new method can effectively compensate for the error of angle measurement in the circular encoder. The peak-to-peak value of the error of angle measurement was reduced from 239.343” to 11.867”, and the repeatability of the calibration results of the new method was less than 2.77”.

## 1. Introduction

The circular encoder is the most commonly used high-precision angular displacement sensor, widely used in various precision instruments and equipment, such as gear measurement centers, laser trackers, and 3D laser scanners [1,2]. In their use, circular encoders are inevitably affected by factors such as their own uneven scribing, installation eccentricity, and environmental changes, resulting in the errors of angle measurement [3,4,5]. To improve the accuracy of angle measurement, it is necessary to calibrate and eliminate the error of angle measurement for the circular encoder [6,7,8].

At present, commonly used calibration methods are divided into two categories: comparison calibration and self-calibration. Comparison calibration is based on angle standards with higher accuracy to measure the error of the circular encoder under calibration, and the accuracy of this method is limited by external reference instruments [9,10,11]. Self-calibration methods do not require external reference instruments, and the calibration accuracy that can be achieved is not limited by the reference instrument.

Nowadays, many self-calibration methods for circular encoders have been developed.

Most circular encoder self-calibration methods use two or more read heads. Probst et al. [12,13,14] developed an angle comparator that consisted of a circular encoder and 16 read heads, where 8 read heads were uniformly distributed and 8 read heads were arranged in pairs. In that system, a new self-calibration method for the indexing plate was adopted to calibrate the circular encoder. Ai C-G et al. [15] investigated the error in angle measurement caused by the eccentricity of circular encoder mounting and calibrated the error of angle measurement using double read heads. Tang S et al. [16] analyzed the impact of installation eccentricity on the accuracy of angle measurement in the circular encoder and proposed a compensation method for installing two read heads in opposite diameter directions. Wang Y-J et al. [17] analyzed the relationship between the installation eccentricity of a circular encoder and the error of angle measurement, established a theoretical model of the error of angle measurement for circular encoder eccentricity based on dual read heads, and proposed a correction method based on the non-radial installation of the dual read heads. Watanabe et al. [18,19] developed the SelfA rotary encoder, consisting of a circular encoder and several sets of sensors. Yi-Tsung et al. [20] proposed a new method to analyze the circular positioning error of a rotary stage based on Abbe’s principle. Nobuyuki Ishii et al. [21] developed a self-calibration method called the Virtual Equal Division Average (VEDA) method that detects higher-order error components of angle measurement systems with fewer read heads. Jiao Y et al. [22,23,24] proposed a four-scan-head circular encoder self-calibration error separation technique based on the Fourier method, which can separate the indexing error of the circular encoder from the influence of the radial error motion of the rotation axis, and achieve the optimal arrangement of four read heads to avoid the suppression of Fourier components and reduce the propagation of errors.

The above methods can effectively compensate for the error of angle measurement and realize the self-calibration of the circular encoder, but they mainly use two or more read heads to realize the self-calibration of the circular encoder, and the calibration accuracy is more dependent on the accuracy of the installation position of the multiple read heads.

There are also some circular encoder self-calibration methods that use a single read head. Orton [25] proposed the spatiotemporal transformation method, which directly converts the rotation time of the spindle to the actual rotation angle of the spindle circular encoder by keeping the spindle at a constant speed, but controlling the spindle speed accurately enough is impractical. Lu X-D [26] proposed a time-measurement dynamic reversal (TDR) method for fast self-calibration of single-read-head circular encoders based on encoder pulse width measurements and free response precision spindle dynamics. However, that method fails to describe the dynamic process of the air-bearing spindle over a wide range of speed variations and only seeks to calibrate accurately within one revolution. Li G-L et al. [27] used harmonic theory to analyze the distribution of eccentricity, tilt, and other order errors in the frequency spectrum and used the harmonic components corresponding to the eccentricity and tilt of the circular encoder as error compensation terms. Based on this, Fourier transform was applied to the multiple measurement results of a single read head to obtain the amplitude values of the first- and second-order error components, and harmonic compensation was then applied to the angle measurement errors of the circular encoder. However, that method cannot effectively eliminate the higher-order components of the angle measurement error of the circular encoder. Sun S-Z et al. [28] proposed a self-calibration method based on the error phase shift (EPS), based on the principle of circumferential space closure, taking the starting measured value sequence as the error shift reference, transforming the error sequence at equal intervals, and obtaining the functional relationship between the error and the reference. However, that method requires uniform motion of the object to be measured. 

To address the shortcomings of the existing methods, this article proposes a new self-calibration method for circular encoders based on inertia and a single read head. Based on the accurate mathematical description of the dynamics of a freely rotating air-bearing spindle under inertia, the method constructs the equation between spindle speed and time across a wide range of speeds during free rotation. Then, the angle interval is divided according to the number of pulses output per revolution of the read head, and the rotation time of the spindle in each angle interval is measured when the spindle rotates freely. Subsequently, the mathematical model of the method is constructed to obtain the proportional relationship between the theoretical value and the actual value of each angle interval, to realize the accurate calibration of the circular encoder over multiple turns and to improve the stability of the self-calibration results.

The remainder of the article is organized as follows: Section 2 explains the relationship between the time intervals and the angle intervals, constructs a mathematical model for the method used in this article, and proposes an iterative method to improve self-calibration accuracy. Section 3 describes the designed experimental systems and analyzes the validity, repeatability, correctness, and superiority of this method based on various experiments. Section 4 contains the summary and outlook.

## 2. Method

### 2.1. Definition of Time Interval and Angle Interval

Encoders are widely used and of various types. This article focuses on the most widely used incremental circular encoders.

Figure 1 shows the correspondence between the time interval and angle interval of the circular encoder pulse. As shown in Figure 1, the span between the adjacent scribes on the circular encoder is recorded as the angular interval $\Delta_{k}$. The time difference, i.e., the time interval $T_{k}$, can be obtained from the moments before and after the rising edges of the two adjacent pulses. Individual intervals are numbered using the rising edge of the zero pulse as the starting point, with the number used as a subscript for intervals, starting from 1.

Ideally, the angle interval corresponding to each counting pulse of A and B after performing quadruple frequency is uniform. The theoretical value of each angle interval is denoted as $\Delta_{0}=2\pi /N$, where *N* is the total number of samples per cycle. However, in practical situations, there is a deviation between the actual value $\Delta_{k}$ and theoretical value $\Delta_{0}$ of each angle interval.

In this article, a dedicated sampling device is presented, designed to accurately record the moment of occurrence of the rising edge of each pulse during the clockwise rotation of the circular encoder, thus obtaining the value of each time interval $T_{k}$. The time interval $T_{k}$ and the actual value $\Delta_{k}$ of the angle interval correspond one-to-one.

The values of $T_{k}$ and $\Delta_{k}$ relate to the spindle speed, as follows:(1)$$\Delta_{k}=\omega_{k}\cdot T_{k}$$
where $\omega_{k}$ is the average angular velocity of the circular encoder during the period $T_{k}$.

### 2.2. Self-Calibration Method

For self-calibration of the circular encoder, an air-bearing spindle with low friction and good axial stability should be used, and the spindle left to rotate freely under inertia with a certain initial angular velocity. The spindle speed gradually decreases due to such factors as air resistance. The data for each time interval are recorded as the spindle gradually decelerates. Self-calibration of the circular encoder can be achieved by analyzing and calculating these data.

The literature [13] suggests that during free rotation of the spindle due to inertia and air resistance, the variation of spindle speed follows a logarithmic law. However, the nonlinear influencing factors existing in the actual system, such as system damping, etc., cause the actual variation law of spindle speed to be inconsistent with the logarithmic law. For more generality, it is determined that the variation of spindle speed within a wide speed range conforms to a polynomial law when the air-bearing spindle is moving inertially and rotating freely, i.e.,:(2)$$\omega (t)=C^{0}+C^{1}t+\ldots +C^{n}t^{n}$$
where ω is the spindle angular velocity, t is the time, n is the polynomial order, and $C^{0},C^{1},\ldots ,C^{n}$ is the polynomial coefficient.

As can be seen, the relationship between the theoretical value $\Delta_{0}$ and the actual value $\Delta_{k}$ of the angle interval is as follows:(3)$$\Delta_{k}=\Delta_{0}+err_{k},\;\;k=1,2,3,\ldots$$
where $err_{k}$ is the difference between the actual value $\Delta_{k}$ and the theoretical value $\Delta_{0}$ of the kth angle interval, i.e., $err_{k}=\Delta_{k}-\Delta_{0}$.

Due to the short time of each pulse interval, the spindle speed varies very little. Using $T_{k}$ and $\Delta_{k}$ to estimate the spindle speed at the midpoint of the kth time interval, i.e., the spindle speed at the moment $S_{k}$ is as follows:(4)$$\omega (S_{k})=\frac{\Delta_{k}}{T_{k}}$$
where $S_{k}=\frac{T_{k}}{2}+\sum_{i=1}^{k-1}T_{i},(k=1,2,3,\ldots )$ denotes the midpoint moment of the kth time interval.

A mathematical model of the calibration method is constructed based on the previous notations and definitions. Via the principle of circle closure and the periodicity of angle intervals, it can be inferred: (5)$$\left\{\begin{array}{l} \Delta_{1}=\Delta_{1+N}=\ldots =\Delta_{1+(m-1)N}\\ \Delta_{2}=\Delta_{2+N}=\ldots =\Delta_{2+(m-1)N}\\ \;\;\;\;\;\;\;\;\;\;\;\;\;\vdots \\ \Delta_{N}=\Delta_{2N}=\ldots =\Delta_{mN} \end{array}\right.$$
where $m=\left\lfloor k/N\right\rfloor$, m is equal to k/N rounded downwards, and m≥n+1.

Combining (3) and (5), the angular velocity $\frac{\Delta_{0}}{T_{1}},\frac{\Delta_{0}}{T_{1+N}},\ldots ,\frac{\Delta_{0}}{T_{1+(m-1)N}}$ can be replaced by $\frac{\Delta_{1}+err_{1}}{T_{1}},\frac{\Delta_{1}+err_{1}}{T_{1+N}},\ldots ,\frac{\Delta_{1}+err_{1}}{T_{1+(m-1)N}}$, i.e.,:(6)$$\;\left\{\begin{array}{l} \frac{\Delta_{0}}{T_{1}}=p_{1}\frac{\Delta_{1}}{T_{1}}\\ \frac{\Delta_{0}}{T_{1+N}}=p_{1}\frac{\Delta_{1}}{T_{1+N}}\\ \;\;\;\;\;\;\vdots \\ \frac{\Delta_{0}}{T_{1+(m-1)N}}=p_{1}\frac{\Delta_{1}}{T_{1+(m-1)N}} \end{array}\right.$$
where $p_{1}=\frac{\Delta_{1}+err_{1}}{\Delta_{1}}=\frac{\Delta_{0}}{\Delta_{1}}$, denotes the proportionality coefficient between the theoretical value $\Delta_{0}$ and the actual value $\Delta_{k}$ of the angle interval.

Substituting (2) and (4) into (6) yields the following:(7)$$\left\{\begin{array}{l} \frac{\Delta_{0}}{T_{1}}=p_{1}\omega (S_{1})=p_{1}(C^{0}+C^{1}S_{1}+\ldots +C^{n}{S_{1}}^{n})\\ \frac{\Delta_{0}}{T_{1+N}}=p_{1}\omega (S_{1+N})=p_{1}(C^{0}+C^{1}S_{1+N}+\ldots +C^{n}{S_{1+N}}^{n})\\ \;\;\;\;\;\;\;\;\;\;\;\;\;\;\;\;\;\;\;\;\;\;\vdots \\ \frac{\Delta_{0}}{T_{1+(m-1)N}}=p_{1}\omega (S_{1+(m-1)N})=p_{1}(C^{0}+C^{1}S_{1+(m-1)N}+\ldots +C^{n}{S_{1+(m-1)N}}^{n}) \end{array}\right.$$

Let $\omega_{1}(t)=p_{1}\omega (t)=C_{1}^{0}+C_{1}^{1}t+\ldots +C_{1}^{n}t^{n}$, where $C_{1}^{0}=C^{0}p_{1}$, $C_{1}^{1}=C^{1}p_{1}$,…, $C_{1}^{n}=C^{n}p_{1}$, as only t is a variable in $\omega_{1}(t)$, use $\left(S_{1},\frac{\Delta_{0}}{T_{1}}\right)$, $\left(S_{1+N},\frac{\Delta_{0}}{T_{1+N}}\right)$,…, $\left(S_{1+(m-1)N},\frac{\Delta_{0}}{T_{1+(m-1)N}}\right)$ to perform least squares fitting on $\omega_{1}(t)$ to obtain a set of parameters ${C_{1}^{0}}^{\prime },{C_{1}^{1}}^{\prime },\ldots ,{C_{1}^{n}}^{\prime }$.

Through the principle of circle closure, we can know $\sum_{i=1}^{mN}\Delta_{i}=2\pi m$, substituting it into (4) to obtain $\sum_{i=1}^{mN}\omega (S_{i})T_{i}=2\pi m$, so $\sum_{i=1}^{mN}\omega_{1}(S_{i})T_{i}=\sum_{i=1}^{mN}{p_{1}}^{\prime }\omega (S_{i})T_{i}={p_{1}}^{\prime }*2\pi m$. Since the parameters of $\omega_{1}(t)$ have been solved by fitting, ${p_{1}}^{\prime }$ can be obtained: ${p_{1}}^{\prime }=\frac{\sum_{i=1}^{mN}\omega_{1}(S_{i})T_{i}}{2\pi m}$.

Similarly, using $\left\{\left(S_{2},\frac{\Delta_{0}}{T_{1}}),(S_{2+N},\frac{\Delta_{0}}{T_{2+N}}),\ldots ,(S_{2+(m-1)N},\frac{\Delta_{0}}{T_{2+(m-1)N}}\right)\right\}$,…, $\left\{\left(S_{N},\frac{\Delta_{0}}{T_{N}}),(S_{2N},\frac{\Delta_{0}}{T_{2N}}),\ldots ,(S_{mN},\frac{\Delta_{0}}{T_{mN}}\right)\right\}$ to solve the above steps, respectively, can obtain ${p_{2}}^{\prime },{p_{3}}^{\prime },\ldots ,{p_{N}}^{\prime }$, and then, according to (6), the following can be obtained:(8)$${\Delta_{j}}^{\prime }=\frac{\Delta_{0}}{{p_{j}}^{\prime }}$$
where 1≤j≤N, ${\Delta_{j}}^{\prime }$ denotes the measured value of the jth angle interval obtained via solving (8). Thereby, ${\Delta_{1}}^{\prime },{\Delta_{2}}^{\prime },\ldots ,{\Delta_{N}}^{\prime }$ can be obtained, and finally, removing the DC component from ${\Delta_{1}}^{\prime },{\Delta_{2}}^{\prime },\ldots ,{\Delta_{N}}^{\prime }$, we obtain the following:(9)$$\left\{\begin{array}{l} {\Delta_{1}}^{\prime \prime }=P{\Delta_{1}}^{\prime }\\ {\Delta_{2}}^{\prime \prime }=P{\Delta_{2}}^{\prime }\\ \;\;\;\;\;\;\;\;\;\vdots \\ {\Delta_{N}}^{\prime \prime }=P{\Delta_{N}}^{\prime } \end{array}\right.\;\;,\;\;\;\;\;P=\frac{2\pi }{\sum_{i=1}^{N}{\Delta_{i}}^{\prime }}$$
where ${\Delta_{k}}^{\prime \prime }$ is the measured value of the kth angle interval with the DC component removed.

Solving (3) can obtain the difference $er{r_{k}}^{\prime \prime }$ between ${\Delta_{k}}^{\prime \prime }$ and $\Delta_{0}$, where $er{r_{k}}^{\prime \prime }={\Delta_{k}}^{\prime \prime }-\Delta_{0}$, the accumulation of which can obtain the error curve of angle measurement for the circular encoder; the formula is as follows:(10)$$Err(k\Delta_{0})=\sum_{i=1}^{k}er{r_{k}}^{\prime \prime }$$

### 2.3. Iterative Method for Improving Self-Calibration Accuracy

In the self-calibration method described in Section 2.2, there exists an approximation: in (4), the average velocity of the air-bearing spindle at the kth time interval is taken as the instantaneous velocity of the spindle at the midpoint moment $S_{k}$ of the kth interval. When the error of angle measurement for the circular encoder is unknown, it is necessary to use approximations for self-calibration, but this inevitably introduces some errors into calibration results. Therefore, this article proposes an iterative method to eliminate such errors. Based on the calibration results ${\Delta_{j}}^{\prime \prime }$ finally obtained in Section 2.2, we can achieve a more accurate calculation of the coefficient $C^{0},C^{1},\ldots ,C^{n}$ of ω(t).

In Figure 2, where $S_{k}=\frac{T_{k}}{2}+\sum_{i=1}^{k-1}T_{i}$, the angle interval corresponding to the kth time interval can be expressed in terms of integration, i.e.,:(11)$$\Delta_{k}=\int_{{S_{k-1}}^{\prime }}^{{S_{k}}^{\prime }}\omega (t)dt$$
where ${S_{k}}^{\prime }=\sum_{i=1}^{k}T_{i}$.

Based on ${\Delta_{j}}^{\prime \prime }$ obtained in Section 2.2, combining (5) and (11) yields the following:(12)$$F=\sum_{i=1}^{mN}{\left(\int_{{S_{i-1}}^{\prime }}^{{S_{i}}^{\prime }}\omega (t)dt-{\Delta_{i}}^{\prime \prime }\right)}^{2}$$

Solving the least squares approximation of (12) to obtain a set of parameters ${C^{0}}^{\prime },{C^{1}}^{\prime },\ldots ,{C^{n}}^{\prime }$, subsequently based on (11) to obtain a new ${\Delta_{1}}^{\prime },{\Delta_{2}}^{\prime },\ldots ,{\Delta_{k}}^{\prime }$, which is averaged and DC components are removed to obtain the following:(13)$$\left\{\begin{array}{l} {\Delta_{1}}^{\prime \prime }=\frac{P}{m}\sum_{i=0}^{m-1}{\Delta_{1+iN}}^{\prime }\\ {\Delta_{2}}^{\prime \prime }=\frac{P}{m}\sum_{i=0}^{m-1}{\Delta_{2+iN}}^{\prime }\\ \;\;\;\;\;\;\;\;\;\vdots \\ {\Delta_{N}}^{\prime \prime }=\frac{P}{m}\sum_{i=0}^{m-1}{\Delta_{N+iN}}^{\prime } \end{array}\right.,P=\frac{2\pi m}{\sum_{i=1}^{mN}{\Delta_{i}}^{\prime }}$$

In this way, the error curve of angle measurement of the circular encoder can be obtained with enhanced accuracy.

On this basis, the calibration accuracy can be further improved by iteration. Based on the ${\Delta_{k}}^{\prime \prime }$ obtained from the self-calibration method in Section 2.2, using the mathematical model proposed in this section to calculate the new ${\Delta_{k}}^{\prime \prime }$ and $\omega^{\prime }(t)$, the obtained new ${\Delta_{k}}^{\prime \prime }$ is then substituted into the mathematical model of this section to obtain the newly calculated results. Subsequently, according to the previous step, repeatedly enter the loop for many iterations (two iterations can achieve significant results) until the difference between this iteration and the previous iteration of the result is less than the set parameter; the end of the iteration is reached and the final result extracted.

## 3. Experiments

### 3.1. Experimental Systems

To validate the self-calibration method of the circular encoder in this article, a set of experimental systems was built, which included the mechanical structure and data acquisition circuit. The mechanical structure of the system is shown in Figure 3, where (a) is a 3D design drawing and (b) is a physical picture. In addition, (b) contains two read heads, but only one read head was used for experiments in this article. The mechanical structure of the system consisted of the RESM20 circular encoder of the Renishaw VIONiC series (52 mm diameter, ±3.97” scribing accuracy, 20 μm pitch, 8192 lines) [29], a V2CLI30D50B read head of the Renishaw VIONiC series (20 nm resolution, clocked at 50 MHz) [30], the circular encoder mounting stage, and a precision air-bearing rotary stage. During the experiments, a force was first applied to the air-bearing rotary stage to make it rotate while carrying the circular encoder. To eliminate the influence of motor torque and friction on the spindle speed, after the rotary stage spindle reached a certain speed, the motor and spindle were separated and the external force was removed, so that the rotary stage was free to rotate under inertial action. During free rotation of the rotary stage, the data acquisition circuit was used to collect time data.

The core chip used in the data acquisition circuit of the experimental system was STM32F103C8T6 (the clock frequency was 72 MHZ) [31]. The Renishaw circular encoder used for the experiments output 8,192,000 pulses per revolution, with each pulse corresponding to a 20 nm circular encoder arc length. If the circular encoder rotates at 1 r/s, the angle resolution corresponding to each clock pulse captured by this data acquisition circuit is 0.018” and the arc length resolution corresponding to the circular encoder is 2.27 nm.

During the experiments, an AM26LS32 [32] chip from Texas Instruments was used as a differential line receiver to receive the output signal of the circular encoder, and the received differential signal was converte into a single-ended signal and sent to the STM32 chip. The STM32 chip recorded the moment when the rising edge of each counting pulse appeared, calculated the time interval corresponding to each counting pulse, and then uploaded the data of each time interval to the upper computer through serial communication for storage.

In addition, a comparative experimental device was built to verify the validity of the method proposed in this article, as shown in Figure 4. In this device, a Renishaw XL-80 laser interferometer (0.001 μm resolution) [33] and a Renishaw XR-20W angle measuring instrument (±1” measuring accuracy) [34] have been added to the original experimental platform.

### 3.2. Experimental Results

The self-calibration experiments were performed using the experimental system described above. During the experiments, the number of samples per revolution of the circular encoder was *N* = 1000, and the time interval $T_{k}$ for each sampling event was captured with a 72 MHz clock rate. A set of time data plots for variations of time interval $T_{k}$ obtained from the experiments are shown in Figure 5. The ripples with periodic fluctuations on the graph reflect the error of angle measurement present for the circular encoder, which is in line with periodic variations of time intervals expected according to the theory.

In Equation (7), taking n=4 and m=10, the data in Figure 5 can be processed using the self-calibration method in Section 2.2 to obtain the ratio of the theoretical value $\Delta_{0}$ and the measured value ${\Delta_{k}}^{\prime }$ of the angle interval, as shown in Figure 6. The measured value ${\Delta_{k}}^{\prime }$ of each angle interval was subsequently obtained through dividing $\Delta_{0}$ (1296”) by the scale factor in Figure 6, and the DC component contained therein was removed to obtain a more accurate measured value ${\Delta_{k}}^{\prime \prime }$ of the angle interval, as shown in Figure 7.

Figure 8 shows the difference $er{r_{k}}^{\prime \prime }$ between the measured value ${\Delta_{k}}^{\prime \prime }$ and theoretical value $\Delta_{0}$ for each angle interval after removing the DC component. The values of the vertical axis in Figure 6, Figure 7 and Figure 8 generally vary in a sinusoidal pattern as a whole with the sampling pulse number, as expected. The curves that move up and down in a small area in the figure mainly reflect the scribing errors that exist in each angle interval of the circular encoder.

To verify the validity of the self-calibration method in this article, the comparative experimental setup shown in Figure 4 was used to measure and calibrate errors of the circular encoder in the experiment, comparing these with the results obtained from the self-calibration method in this article. As shown in Figure 9, the error curve of angle measurement is obtained by accumulating $er{r_{k}}^{\prime \prime }$ in Figure 8 according to Equation (10), with peaks of 229.993” and −9.35” and an error of 239.343”; the residual error curve is obtained by the error obtained from the method in this article minus the error measured with the Renishaw XL-80 laser interferometer, with peaks of 1.266” and −10.601” and an error of 11.867”. In addition, the residual error curves in Figure 9 do not overlap the values at the 0° and 360° positions (there is a deviation of 2.59”), which is due to the limited experimental conditions; the ambient temperature and the ambient vibration were not controlled well enough, leading to the non-closure of the results measured with the laser interferometer [35]. However, the trend of the laser interferometer results shows that the method described in this article is correct, albeit that the accuracy of the experiment can be improved again. The experimental results show that, under the condition of not relying on external reference, the self-calibration method presented in this article can better compensate for the error of angle measurement of the circular encoder and improve the accuracy of angle measurement for circular encoder measurement systems.

### 3.3. Repeatability of Experimental Results

Take n=2 and m=5, and substitute the data from different regions in Figure 4 into the self-calibration method in this article. Use the difference curve of the results obtained from different data to reflect the repeatability of the method in this article. As shown in Figure 10, the difference in results obtained using data from different laps in the same experiment is less than 1.17”.

To investigate the effect of the initial spindle speed on the experimental results, this study included experiments in which the spindle rotated freely at different initial speeds. The curves in Figure 11 represent the difference curves of the results obtained from the experiments with different initial velocities, which fluctuated within a range of 2.77”. Figure 10 and Figure 11 reflect that the results obtained with the present method have good reproducibility.

### 3.4. Comparative Experiments

To better verify the calibration effect of the self-calibration method in this article, the TDR method described in the literature [26] was also reproduced and compared with the method proposed in this article. The data in Figure 4 were substituted into the method proposed in this article and the TDR method, respectively, to obtain two results. Then, the results of the TDR method were subtracted from the results obtained in this article to obtain a difference curve. As shown in Figure 12, the difference between the results obtained with this article’s method and the TDR method is small, which further illustrates the correctness of this article’s method. The repeatability of the results obtained with the TDR method is shown in Figure 13, with a difference fluctuation range of 5.11“, which is larger than the repeatability of the method in this article of 2.77”. It is thereby evident that the method presented in this article has better calibration repeatability for data with a larger range of speed change, reflecting the superiority of the self-calibration method in this article.

### 3.5. Experiments of Iterative Method

To verify the effectiveness of the iterative method for improving calibration accuracy proposed in Section 2.3, a set of data was processed using the self-calibration method. Where n=4 and m=5, and the repeatability curves of the results obtained before and after the addition of iterations are shown in Figure 14. Figure 14a shows the repeatability of the results before iteration, with a fluctuation range of 2.72”, and Figure 14b shows the repeatability after the second iteration, with a fluctuation range of 1.099”. As can be seen, the iterative method proposed in Section 2.3 can effectively improve the reproducibility of the calibration results when the sampling data are few and the polynomial order *n* is large.

However, the iterative method proposed in Section 2.3 is more computationally intensive and time-consuming, and the effect is not significant when the polynomial order is low. For example, when n=2 and m=5, the repeatability of the results obtained before and after iterations changes very little; this difference was less than 0.0855” in the experiment, and the computing time was about 74 s.

## 4. Conclusions

Aiming at the addressing the shortcomings of the existing circular encoder self-calibration methods, this article proposes a new self-calibration method for circular encoders based on inertia and a single read head. The method proposed in this article uses a polynomial to fit the velocity curve. When applied to a wide range of velocity variations during free rotation, the consistency of the results of the self-calibration for the error of angle measurement of the circular encoder is better than those of the traditional method. Compared with the multiple-read-head method, this method can reduce the cost of the angular measurement system and ensure the calibration accuracy.

The conclusions of the research work in this article are as follows:A new mathematical model for the self-calibration method for a circular encoder based on inertia and a single read head was developed, and the experimental system for the new method was constructed for experimental verification. The experimental results show that the new method can quickly and effectively compensate for the error of angle measurement in circular encoder angle measurement systems without relying on a high-precision reference apparatus. The peak-to-peak value of the error of angle measurement was reduced from 239.343” to 11.867”, indicating a significant improvement in angle measurement accuracy. The repeatability of the results obtained with the new method was less than 2.77”.In addition, a replication experiment using the TDR method was conducted to compare it with the method in this article. The results showed that the difference was small, with fluctuations within 2.266”. For the same data, the repeatability of the TDR method was 5.11”, while the repeatability of the method in this article was within 2.77”. These findings further prove the correctness and superiority of the method in this article.An iterative method that can further improve the calibration accuracy is also proposed and was experimentally verified. The experimental results show that it reduced the repeatability of the calibration results from 2.72” to 1.099” when the sampling data were few and the polynomial order *n* was taken to be large, indicating the effectiveness of the iterative method. However, the iterative method is more computationally intensive and time-consuming, and the effect is not significant when the polynomial order is low. Therefore, it is reasonable not to use this iterative method for processing when the requirements for calibration accuracy are not very high.

Due to the limitations of the conditions, the experimental setup in this article cannot be used to carry out experiments with higher initial angular speeds of the spindle. Furthermore, the clock frequency of the acquisition device was low, and the total number of samples *N* per revolution was correspondingly low. In subsequent research, the total number of samples *N* per revolution will be significantly increased, and FPGA will be used to collect time interval data when the initial angular velocity of the spindle is higher. These measures are expected to further improve the calibration accuracy and the repeatability of the method proposed in this article.

## Figures and Tables

**Figure 1 sensors-24-03069-f001:**
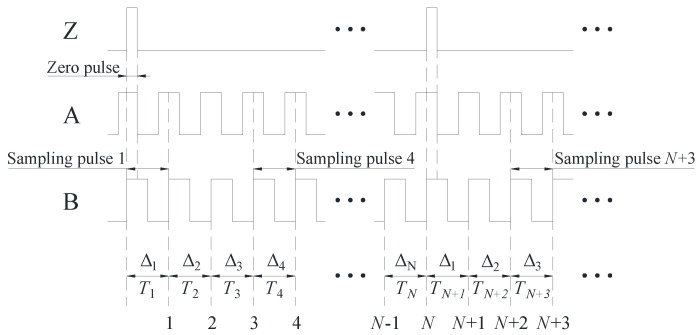
Correspondence between the time interval and angle interval of pulse for the circular encoder.

**Figure 2 sensors-24-03069-f002:**
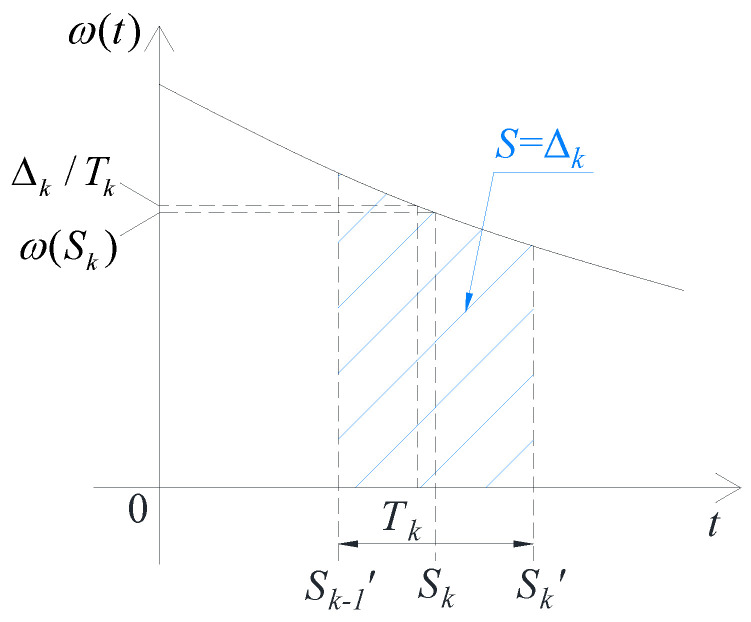
Principle of the method for improving self-calibration accuracy.

**Figure 3 sensors-24-03069-f003:**
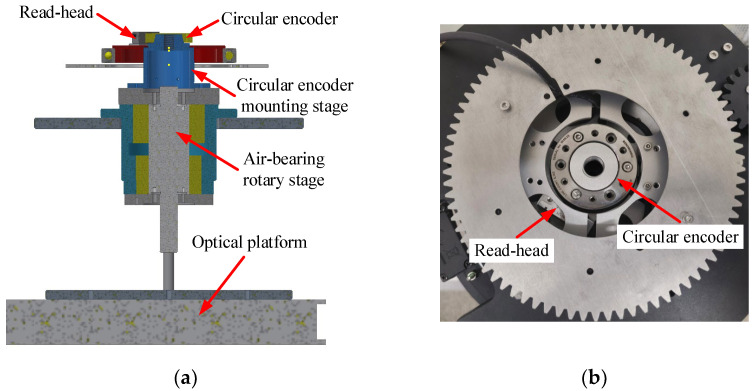
Structure of experimental system. (**a**) 3D view of experimental system. (**b**) Physical device—top view.

**Figure 4 sensors-24-03069-f004:**
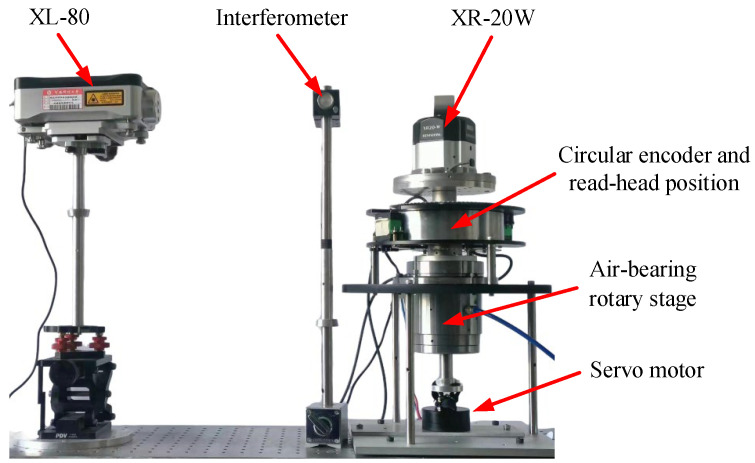
Comparison experiment setup using Renishaw XL-80 laser interferometer.

**Figure 5 sensors-24-03069-f005:**
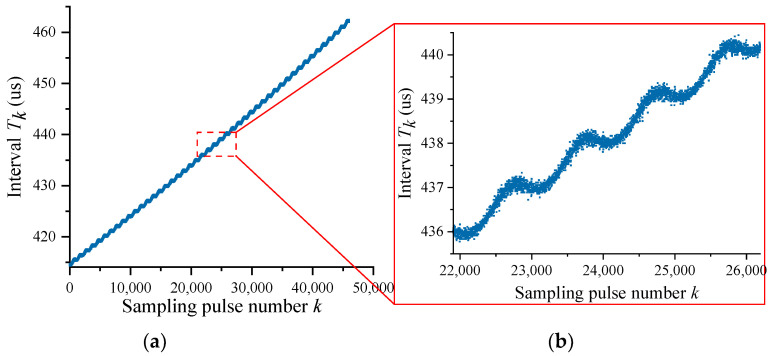
Relations between sampling pulse numbers and time intervals: (**a**) overall distribution; (**b**) zoomed in.

**Figure 6 sensors-24-03069-f006:**
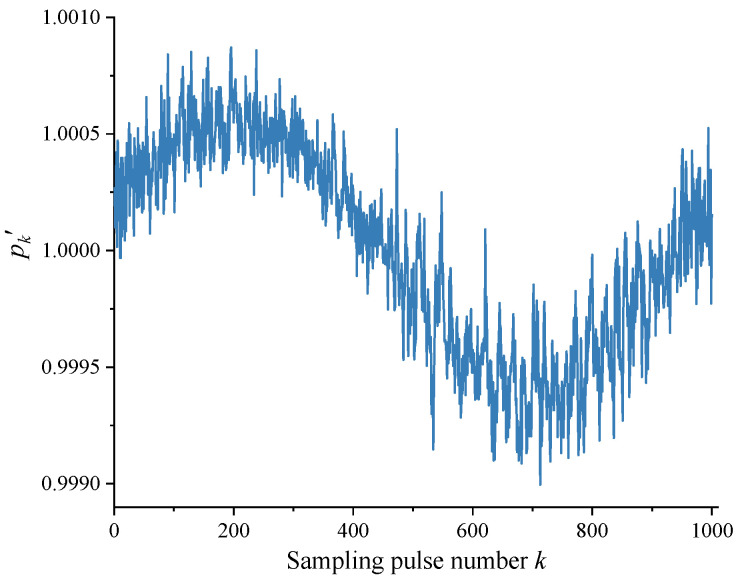
Ratio of theoretical value $\Delta_{0}$ to measured value ${\Delta_{k}}^{\prime }$ for angle intervals.

**Figure 7 sensors-24-03069-f007:**
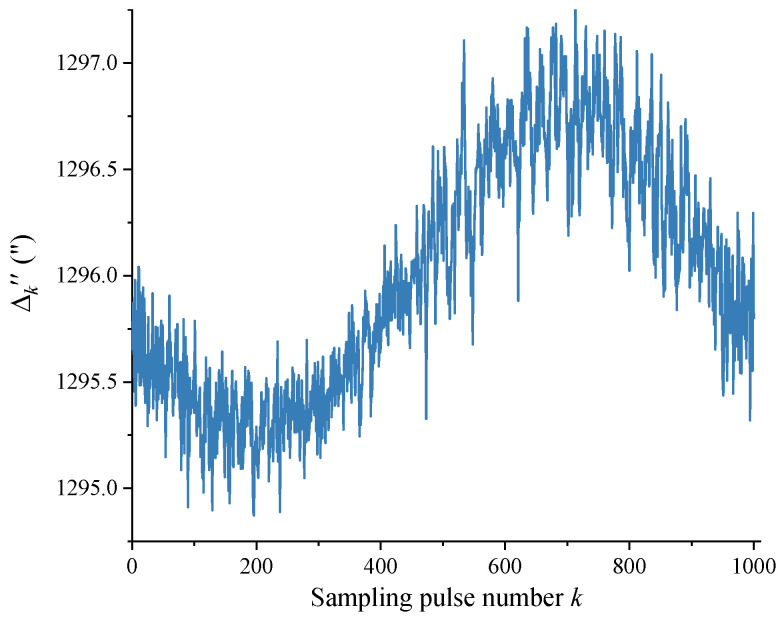
Measured value ${\Delta_{k}}^{\prime \prime }$ for each angle interval.

**Figure 8 sensors-24-03069-f008:**
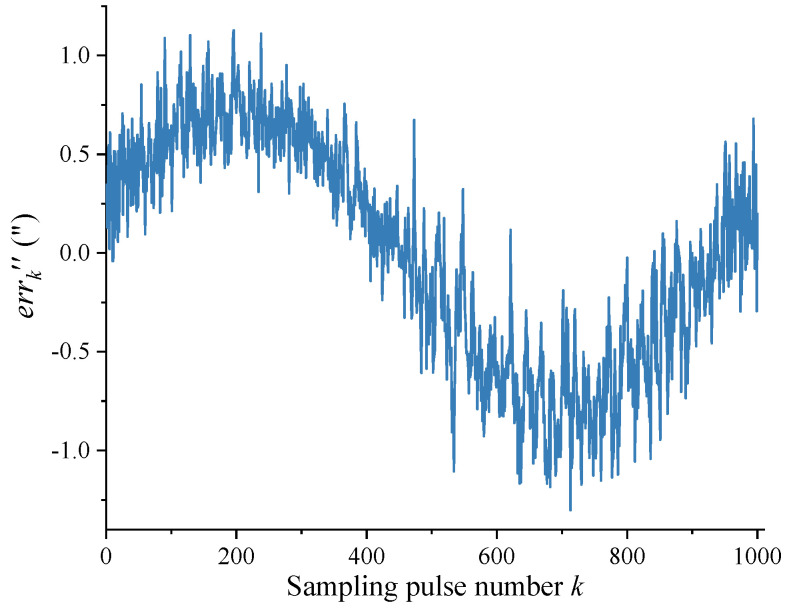
Error of angle measurement for each angle interval.

**Figure 9 sensors-24-03069-f009:**
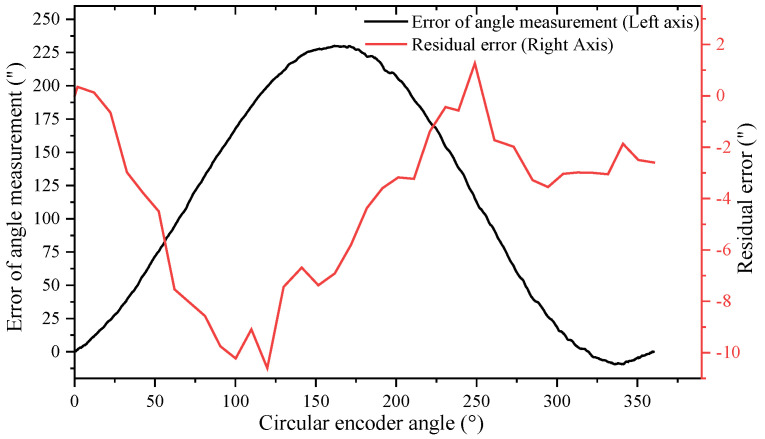
Self-calibration error curve and residual error curve.

**Figure 10 sensors-24-03069-f010:**
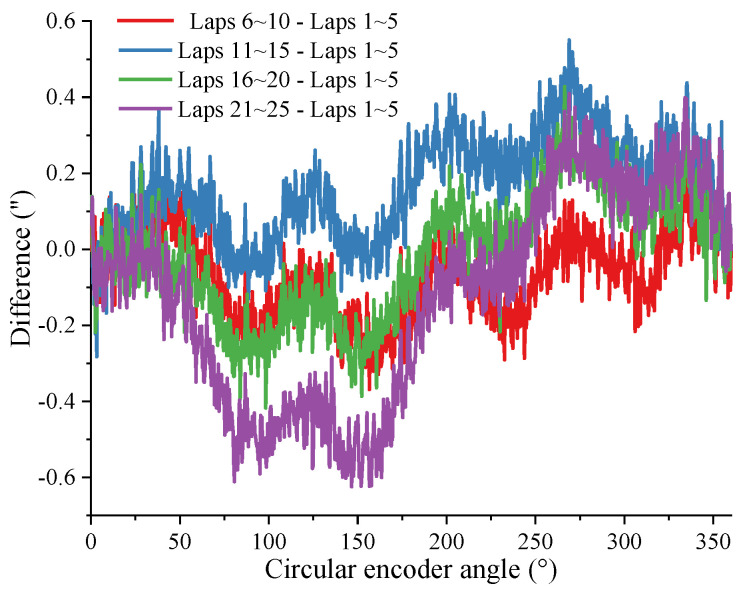
Repeatability of results obtained from different data in the same experiment.

**Figure 11 sensors-24-03069-f011:**
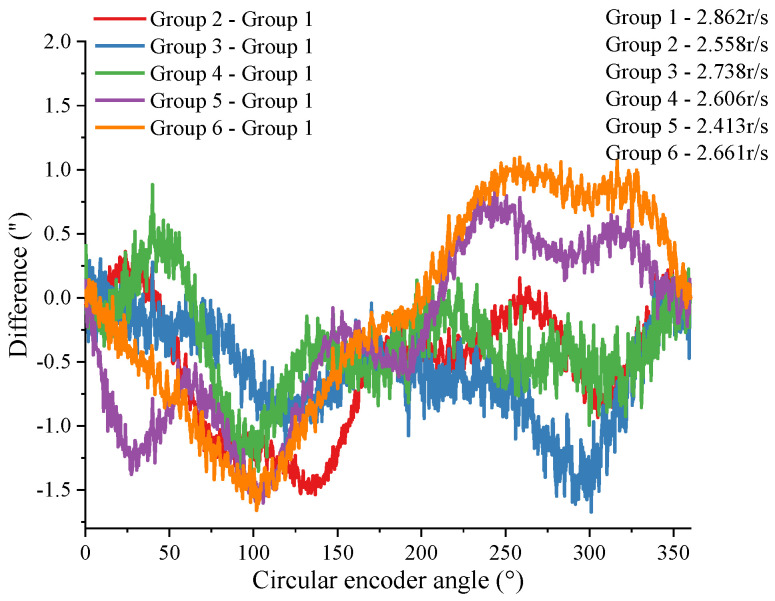
Repeatability of different initial velocities under the proposed method.

**Figure 12 sensors-24-03069-f012:**
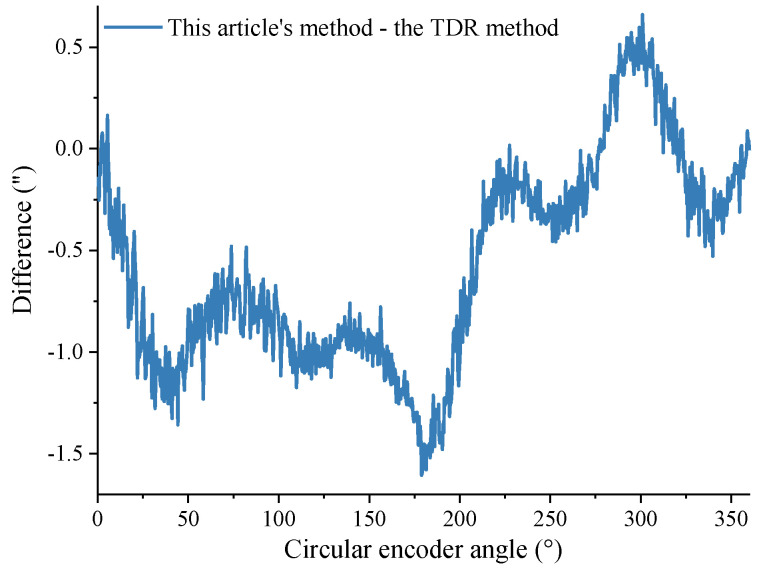
Difference in the results between proposed method and TDR method.

**Figure 13 sensors-24-03069-f013:**
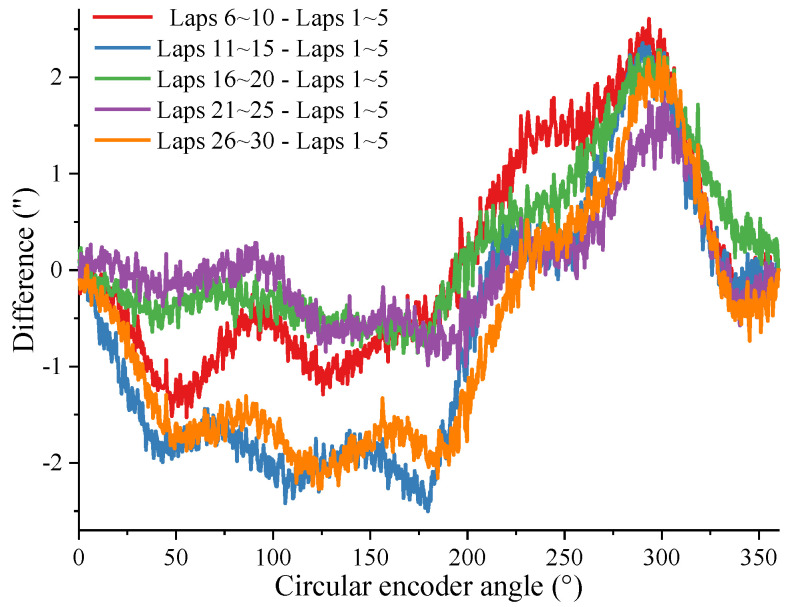
Repeatability of the TDR method.

**Figure 14 sensors-24-03069-f014:**
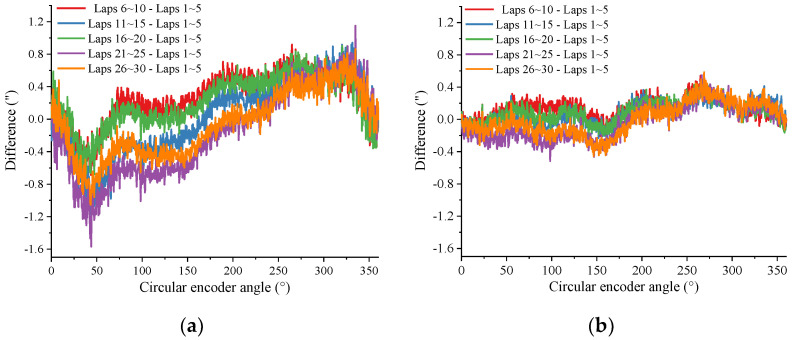
Comparison of repeatability before and after iteration: (**a**) before iteration. (**b**) after 2 iterations.

## Data Availability

The data that support the findings of this study are available from the corresponding author upon reasonable request.

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
