# Peer review of "Self-Calibration Method for Circular Encoders Based on Inertia and a Single Read-Head"

_sensors, 2024, doi:10.3390/s24103069_

Round 1
Reviewer 1 Report
Comments and Suggestions for Authors
Thwу results are very interesting and definitely are to be published. However, to our opinion, the manuscript has yet to be improved. We have not revealed any mistakes or flaws in the manuscript, but some places, to our opinion, are to be additionally clarified for the reader’s convenience.
1. Various people have rather different idea of what is the angle sensor and what is encoder. Somу people are working with incremental encoders, some – with the absolute ones, and some with emulations of encoder (for instance by ring laser etc.). Some specialists even mix angle sensors with other devices – e.g. gyroscopes etc. Hence to our opinion it is necessary to start the section 2 (just after the introduction) by a brief explanation of the device under consideration, principle of its operation and some comments of the encoders (we guess that the method is applicable only to devices with the simplest uniform circular incremental encoder?).
2. On the basis of such introduction it would be much simpler eo explain the correspondence between the time interval and angle interval of pulse for the circular en-coder.(Fig.1). In the current version of the manuscript it is a little bit difficult for understanding for the newcomer
3. Notation “±3.97" scribing accuracy.” (page 4) – even if it taken from some manufacturer’s documents – sounds strange. It should be, say ±4" .
4. The nature and origin of the curves in the Fig.4 require more detailed explanation.
5. The photograph, shown in the Fig.8, is absolutely non-informative. To our opinion it should be replaced (or accompanied) by the optical scheme of the comparison experiment.
Comments on the Quality of English Language
English language of the manuscript is clear and readable. However, we reccomend the authors to replace some very long complicated sentences by several shorter and simpler ones,
Reviewer 2 Report
Comments and Suggestions for Authors
In the article, its authors described the new and original method of self-calibration of circular encoders. The presented results relate to research work in the field of applied sciences. Described research can have great practical significance. In particular: the authors described the mathematical model of the method, then they discussed how to implement the new method and they included the results of validation tests. Experimental results confirmed that the new method achieves higher calibration accuracy than in other cases described in the literature, which concern similar mid-range technical solutions. Thus, the own contribution of the authors of the paper to the development of the field has been demonstrated, which allows the publication of the paper in the form presented - in terms of substantive scope.
Used methodology and testing procedure are described in a correct and substantial way. In the article, the equations and figures provided are clear and described in an understandable way.
The weaker side of the paper is the way of interpreting the obtained results, especially the content given in Conclusions. Conclusions should not be a summary of the conducted research but a critical evaluation of the obtained results and a comparison with the state of not only knowledge but also technology.
In view of the fact that the described work may have great practical importance, in Conclusions it would be necessary to compare the results not only with the results of other publications - but also with practically applied solutions. Therefore, I recommend supplementing the article in this field.
I also suggest giving the website where Renishaw's VIONiC reports detailed technical parameters of the V2CLI30D50B head, which was used during the experiment. The reader should be able to verify the results described.
The text of the paper needs a minor editorial correction, as I write about in the section on the language of the article.
Comments on the Quality of English Language
The language of the work is generally understandable, although there are some excessively long sentences in the text, such as the second sentence in Conclusions. Overly long sentences can be divided into several shorter ones, then their meaning will be clearer.
Sometimes there is a change in the writing style in the paper, such as in Section 3.5 Experiments of iterative method. Such a change of style is negatively distinguished from the form at description the neighboring chapters.
There are unnecessary shortcuts of thought in the work, which the writers do not see but the reader registers. This occurs, among other things, when the content of the article is identifiable with the research performed, or drawings with physical objects - such as the caption in Figure 8.
Therefore, I recommend, once again read the text of the paper in detail and make corrections to the small stylistic errors noted.
